# Repeated Dose of Contrast Media and the Risk of Contrast-Induced Acute Kidney Injury in a Broad Population of Patients Hospitalized in Cardiology Department

**DOI:** 10.3390/jcm12062166

**Published:** 2023-03-10

**Authors:** Małgorzata Cichoń, Maciej T. Wybraniec, Oliwia Okoń, Marek Zielonka, Sofija Antoniuk, Tomasz Szatan, Katarzyna Mizia-Stec

**Affiliations:** 1First Department of Cardiology, School of Medicine in Katowice, Medical University of Silesia, 40-635 Katowice, Poland; 2European Reference Network on Heart Diseases (ERN GUARD-HEART), 1105 AZ Amsterdam, The Netherlands; 3Department of Cardiology in Cieszyn, Upper-Silesian Medical Center, 40635 Katowice, Poland

**Keywords:** contrast-induced acute kidney injury, CI-AKI, contrast-induced nephropathy, CIN, repeated contrast medium

## Abstract

Contrast-induced acute kidney injury (CI-AKI) can lead to the development of chronic kidney disease (CKD) and impaired in-hospital and long-term outcomes among cardiac patients. The aim of this study was to evaluate the impact of repeated contrast media (CM) administration during a single hospitalization on the rate of CI-AKI. The study group (*n* = 138) comprised patients with different diagnoses who received CM more than once during hospitalization, while the control group (*n* = 153) involved CAD patients subject to a single CM dose. Following propensity score matching (PSM), both groups of *n* = 84 were evenly matched in terms of major baseline variables. CI-AKI was defined by an absolute increase in SCr ≥ 0.3 mg/dL or >50% relative to the baseline value within 48–72 h from the last CM dose. Patients in the study group were older, had a higher prevalence of diabetes and CKD, received a higher total volume of CM, had a lower left ventricular ejection fraction, lower prevalence of multivessel coronary artery disease (MV-CAD), and a trend towards a lower prevalence of arterial hypertension and smoking. SCr did not differ between the study and control groups at 72 h after the CM use. CI-AKI occurred in 18 patients in the study (13.0%) and in 18 patients (11.8%) in the control group (*p* = 0.741). The rate of CI-AKI was also comparable following the PSM (13.1% vs. 13.1%, *p* = 1.0). Logistic regression analysis revealed that CKD, diabetes mellitus, MV-CAD, age, and non-steroidal anti-inflammatory drugs use, but not repeated CM use, were independent predictors of CI-AKI.

## 1. Introduction

Contrast-induced acute kidney injury (CI-AKI) is the third most common cause of new kidney function worsening in hospitalized patients after renal hypoperfusion and postoperative AKI. Due to a massive increase in contrast medium (CM) exposure during the diagnostic procedures of the cardiovascular system, CI-AKI has emerged as a growing problem among cardiac patients, which can lead to the development of chronic kidney disease (CKD) and impaired in-hospital and long-term outcomes.

The direct nephrotoxic effect of CM on tubular epithelial cells and vasoactive molecules induces ischemic renal cell injury by increased oxidative stress [1]. These processes could affect the development of CI-AKI in up to one in four patients undergoing CM administration. Although kidney function worsening is usually transient and the serum creatinine concentration (SCr) returns to baseline within 7–10 days, CI-AKI is associated with serious adverse events, including short- and long-term mortality [2].

Kidney Disease Improving Global Outcomes (KDIGO) recommends assessing the risk for CI-AKI in all patients who are considered for a procedure that requires intravascular administration of iodinated CM including coronary angiography, percutaneous coronary intervention, and computed tomography [3]. Although several novel renal injury biomarkers have been proposed to facilitate early diagnosis of CI-AKI, serial serum creatinine concentration analysis persists as a gold standard of proceeding [4].

Advanced age, diabetes, heart failure, diabetes, hypertension, as well as the amount and type of CM, are well-established risk factors for CI-AKI [5]. On the other hand, using the minimum necessary dose of iso-osmolar or low-osmolar agents, maintenance of hydration status, and avoiding nephrotoxic pharmacotherapy are the most effective interventions to protect against CI-AKI [6,7,8].

Despite the risk of worsening kidney function, many cardiology patients require more than one procedure with CM use within the timeframe of a single hospitalization. The aim of this study was to evaluate the impact of repeated CM administration during the index hospitalization on the rate of CI-AKI defined by an absolute increase in SCr ≥ 0.3 mg/dL or > 50% relative to the baseline value [3].

## 2. Methods

This was a retrospective, single-center study that covered 291 patients admitted to the First Department of Cardiology, Medical University of Silesia, Upper-Silesian Medical Center in Katowice, Poland. The study group (*n* = 138) comprised patients with different diagnoses who received CM more than once during hospitalization (coronary angiography/percutaneous coronary intervention/computed tomography). The control group (*n* = 153) involved coronary artery disease (CAD) patients subjected to a single CM dose. The necessary data were acquired from the database of the hospital, and all the data were fully anonymized before access. No additional approval from the Ethics Committee was required due to the anonymous and non-interventional design of the project.

We retrospectively collected the following variables: demographic data: age, sex, weight, height, body mass index (BMI);volumes of first, second, and third contrast administrations, total volume of administered contrast, and total volume of contrast to body weight ratio; diagnosis and performed procedure: coronary angiography, percutaneous coronary intervention, computed tomography;laboratory blood morphological and biochemical parameters;serum creatinine level: before the procedure, and daily for 3 days after the administration of the contrast agents;echocardiography and electrocardiography parameters;past medical history.

Exclusion criteria involved incomplete data regarding the volume of contrast media and the serum creatinine level. 

Due to the registry design of the study, the research was exempt from full consent by the Ethics Committee of the Medical Universal of Silesia in Katowice, Poland in 2021.

### Statistical Analysis

The distribution of the continuous variables was verified using the Shapiro–Wilk’s test. The continuous variables were expressed as the arithmetic mean ± the standard deviation (SD) or the median and 1–3 quartile boundary, whereas the categorical variables were shown as absolute counts with percentages (%). In the case of the continuous variables, the Mann–Whitney test or the Kruskal–Wallis test was applied, while, in the case of the qualitative parameters, the chi-square test was utilized. Univariable odds ratios with 95% confidence intervals (95%CI) were calculated for the prediction of the presence of CI-AKI. Subsequently, all the parameters with *p* < 0.1 were incorporated into the stepwise multivariable logistic regression model in order to establish the independent predictors of CI-AKI. A receiver operating characteristics curve was plotted for independent predictors of CI-AKI. A propensity score matching (PSM) analysis was performed to compensate for the uneven distribution of baseline variables between the study and the control groups. Both groups were matched using the nearest neighbor algorithm with a 1:1 ratio, taking into consideration the following set of variables: age, sex, left ventricular ejection fraction (LVEF), chronic kidney disease (CKD), estimated glomerular filtration rate (eGFR), the prevalence of arterial hypertension, diabetes mellitus, heart failure, and multivessel coronary artery disease. A two-sided *p*-value of 0.05 was considered statistically significant.

## 3. Results

### 3.1. Demographic and Clinical Characteristics

The baseline demographic and clinical characteristics were highlighted in Table 1. 

The majority of patients were admitted due to non-ST-elevation acute coronary syndrome (NSTE-ACS) (169; 58.1%), followed by ST-segment-elevation myocardial infarction (STEMI) (67; 23.0%), aortic valve stenosis (38; 13.1%), chronic coronary syndrome (13; 4.5%), and aortic aneurysm (3; 1.0%). Moreover, stenosis of >50% in the coronary arteries was present in 223 patients (78.0%), and 100 (34.6%) patients were diagnosed with multivessel coronary artery disease (MV-CAD). Regional wall motion abnormalities were recognized in 200 (69.0%) patients. A total of 71 (24.5%) patients suffered from chronic kidney disease (CKD). The most prevalent drugs in the overall population were statins (61.5%), ACEI/ARB (59.5%), loop diuretics (24.8%), and NSAIDs (10.2%).

The vascular access in the overall group was divided almost equally between radial and femoral, used, respectively, in 131 (48.9%) and 118 (44.0%) patients, while brachial access was used in only 19 patients (7.1%) Table 1.

#### 3.1.1. Study vs. Control Group

The study group consisted of 138 patients with different diagnoses who received CM more than once, while 153 CAD patients after a single CM dose were included in the control group (*n* = 153). The demographic and clinical characteristics of the study and control are presented in Table 1. 

#### 3.1.2. Admission Diagnosis

An admission diagnosis of STEMI, chronic coronary syndrome, and aortic aneurysm were more common in the study group, while NSTE-ACS and aortic valve stenosis were diagnosed more frequently in the control group.

#### 3.1.3. Medical History

In the study, the groups differed in the frequency of CKD (31.4%) in the study group to 18.3% in the control group, *p* = 0.01), while there were no differences in the frequency of a history of acute kidney disease (AKI) and history of nephrectomy. A history of myocardial infarction (MI), percutaneous coronary intervention (PCI), atrial fibrillation, and hyperlipidemia was more frequently observed in the control group, while diabetes mellitus was more often diagnosed in the study group.

The study revealed that patients in the study group were more often taking loop diuretics (32.1% in the study group to 19.6% in the control group, *p* = 0.023). Both statins (42.7% in the study group to 73.7% in the control group, *p* < 0.001) and ACEi/ARB (49.3% in the study group to 68.6% in the control group, *p* = 0.001) were more prevalent in the control group. 

#### 3.1.4. Echocardiography and Laboratory Testing 

A lower left ventricular ejection fraction (LVEF) and higher troponin concentration in the laboratory testing were observed in the study group. There were no differences in the baseline SCr and its change over time. 

#### 3.1.5. Coronary Angiography

During coronary angiography, radial access was the most frequently used vascular access in the study group, whereas, in the control group, femoral access was more prevalent. Stenosis of >50% in coronary arteries was more frequently observed in the study group (91.2% in the study group to 65.8% in the control group, *p* < 0.001), and the patients more often required percutaneous coronary intervention (PCI) (82.5% in the study group to 50% in the control group, *p* < 0.001). The study group received an average of 265 mL of contrast, while the control group was given 120 mL (*p* < 0.001).

#### 3.1.6. Propensity Score Matching: Study vs. Control Group

The study and control groups were matched using PSM in terms of the baseline variables, including age, sex, left ventricular ejection fraction (LVEF), chronic kidney disease (CKD), estimated glomerular filtration rate (eGFR), the prevalence of arterial hypertension, diabetes mellitus, heart failure, and MV-CAD Figure 1. The Hansen and Bowers overall balance test for the model was *p* = 0.927. The analysis yielded *n* = 84 cases in the study and control groups.

The study and control groups differed only in terms of the initial diagnosis and the volume of CM (median of 230 [180; 300] mL vs. 100 [80; 155] mL; *p* < 0.001). After the PSM, the rate of CI-AKI was comparable in the study and control groups (13.1% vs. 13.1%, *p* = 1.0).

#### 3.1.7. Non-CI-AKI vs. CI-AKI Group

In the study, CI-AKI was diagnosed in 36 patients (12.4% of the overall population). The demographic and clinical characteristics of the non-CI-AKI and CI-AKI groups are presented in Table 2.

#### 3.1.8. Admission Diagnosis

While the study revealed no differences in the initial diagnosis of STEMI, NSTE-ACS, chronic coronary syndrome, aortic valve stenosis, or aortic aneurysm, patients in the CI-AKI group more often experienced a sudden cardiac arrest during the index hospitalization (4.62% in non-CI-AKI group to 14.29% in CI-AKI group, *p* = 0.045).

#### 3.1.9. Medical History

The results confirmed that the chances of AKI occurrence are increased by age (66.78 ± 11.49 in the non-CI-AKI group to 73.31 ± 10.80 in the CI-AKI group, *p* = 0.01), diabetes mellitus (30.31% in the non-CI-AKI group to 55.56% in the CI-AKI group, *p* = 0.003), CKD (20.87% in the non-CI-AKI group to 50% in the CI-AKI group, *p* < 0.001), and the history of MI (−27.95% in the non-CI-AKI group to 50% in the CI-AKI group, *p* = 0.007).

The chances of AKI were also higher in patients with a history of AKI (1.97% in the non-CI-AKI group to 8.33% in the CI-AKI group, *p* = 0.029). The study revealed that multivessel coronary artery disease (MV-CAD), as well as peripheral artery disease (PAD), increased the risk of AKI onset (30.71% in the non-CI-AKI group to 62.86% in the CI-AKI group, *p* < 0.001 for MV-CAD and 18.65% in the non-CI-AKI group to 36.11% in the CI-AKI group *p* = 0.016 for PAD).

#### 3.1.10. Echocardiography and Laboratory Testing 

Patients in the non-CI-AKI group are characterized by smaller left atrial dimensions. The study showed that a lower LVEF is associated with a higher risk of CI-AKI (48.46% ± 11.22 in the non-CI-AKI group to 42.28% ± 14.43 in the CI-AKI group, *p* = 0.017). 

In the laboratory testing, a higher sodium concentration was observed in the non-CI-AKI group. There were no differences in the baseline serum creatinine concentration between the non-CI-AKI and CI-AKI groups.

#### 3.1.11. Coronary Angiography

In the coronary angiography, the presence of stenosis of >50% in the coronary arteries was higher in the CI-AKI than in the non-CI-AKI group (94.3% vs. 75.7%, *p* = 0.013). However, in view of the results of our work, repeated doses of CM did not seem to increase the risk of CI-AKI. There were no significant differences in the volume of the first, second, and third contrast administrations, as well as in the total contrast volume.

#### 3.1.12. Predictors of CI-AKI

In the univariable logistic regression analysis, CKD, diabetes, MV-CAD, and PAD significantly increase the risk of AKI (*p* = 0.0062 for CKD, *p* = 0.0267 for diabetes, *p* = 0.004 for MV-CAD, and *p* = 0.0228 for PAD) Table 3.

The study also revealed that therapy with NSAIDs, as well as loop diuretics, had an impact on the risk of AKI development (*p* = 0.0136 for NSAIDs and *p* = 0.0381 for loop diuretics) Table 3.

In the multivariable logistic regression, age, CKD, diabetes, MV-CAD, and NSAIDs use were associated with a higher AKI risk (*p* = 0.0044 for age, *p* = 0.0383 for CKD, *p* = 0.0433 for diabetes, *p* = 0.0005 for MV-CAD, and *p* = 0.0044 for NSAIDs) Table 3. 

The predictive model of the AKI-onset resulted in the area under the receiver operating characteristic (AUROC) curve of 0.910 (*p* = 0.8) Figure 2.

## 4. Discussion

While CI-AKI constitutes a relatively common problem in clinical practice, only a few studies have concerned multiple CM admission. Further research is needed to fully explain the problem of CI-AKI in patients with cardiovascular disease. The main valuable finding of this study is the identification of independent risk factors for CI-AKI in patients who have undergone several procedures with CM usage. 

The study revealed that the patient’s clinical characteristics, but not the CM dose, have an impact on the risk of developing CI-AKI. In the study period, diabetes, CKD, LVEF, MV-CAD, PAD, NSAIDs, and loop diuretics use turned out to be risk factors for kidney function deterioration. This finding is consistent with the results of previous studies concerning this problem. 

In the meta-analysis in which patients with STEMI were analyzed, CI-AKI occurred in 13.3% of patients who underwent PCI [9]. Age (OR: 7.79, 95%CI: 5.24–10.34; *p* < 0.00001), diabetes (OR: 1.83, 95%CI: 1.47–2.29; *p* < 0.00001), and LVEF (OR: −6.15, 95%CI: −9.52–−2.79; *p* = 0.0003), as well as the estimated glomerular filtration rate (eGFR) (OR: 3.12, 95%CI: 2.21–4.40; *p* < 0.00001), showed an association with CI-AKI risk, which is in accordance with our results. In addition, a history of previous MI, left anterior descending artery stenosis, and the presence of acute heart failure also significantly increased CI-AKI frequency [9].

Of note, a CI-AKI predictive model was designed, which incorporated a set of clinical variables [10]. The predictive value of the clinical risk score based on age, LVEF, and eGFR was evaluated in the study by Ando et al. [10]. Multivariate analysis revealed that age, eGFR, LVEF, and TIMI flow grade were independent CI-AKI risk factors. The proposed three-variable risk score, based on age, eGFR, and LVEF, proved to be correlated with CI-AKI frequency (OR: 5.19, *p* < 0.001, AUC 0.88). That finding is compatible with the results of our study, in which age, CKD, and LVEF were associated with CI-AKI risk. In addition, the cited study revealed that in-hospital mortality was higher in patients with kidney function deterioration. 

Diabetes turned out to be one of the CI-AKI predictors in patients with cardiovascular diseases included in our research. In the meta-analysis by Liu et al. [11], including 84 studies and 1,136,827 participants, diabetes was an independent risk factor for CI-AKI (OR: 1.58, 95%CI: 1.48–1.70, I2 = 64%). In addition, the predictive effect of elevated CI-AKI was stronger in patients with coexisting CKD (OR: 2.33, 95%CI: 1.21–4.51). 

In the study by Wybraniec et al. [12], in addition to diabetes, also age, coexisting PAD, and complex MV-CAD were independent predictors of CI-AKI. In addition, the study revealed that intra-renal blood flow parameters, such as the renal resistive index, could predict the development of CI-AKI.

Taking into account the influence of the amount of CM on the risk of developing CI-AKI, the study results are inconsistent. In the study by Mehran et al. [13] carried out on a large contemporary PCI cohort, CI-AKI occurred in 4.3% of the patients. The main independent factors for CI-AKI in this study proved to be eGFR, LVEF, heart failure, diabetes, and age. Moreover, patients who required complex PCI were at high risk of developing CI-AKI. In contrast to our results, the study also revealed that the amount of CM given to the patient increases the risk of CI-AKI.

On the other hand, in a meta-analysis of AKI occurring after computed tomography by Aycock et al. [14], which included 28 studies, contrast-enhanced computed tomography was not associated with higher AKI risk (OR: 0.94, 95%CI: 0.83–1.07). Compared to the group of patients receiving non-contrast CT, the study group did not differ in AKI frequency, need for renal replacement therapy, or all-cause mortality. This finding identifies patient- and illness-level factors rather than the use of CM as a cause of AKI. 

In the meta-analysis by Obed et al. [15] comprising the data of 169,455 patients, computed tomography with CM usage was not associated with a higher risk of AKI (OR: 0.97, 95%CI: 0.85–1.11, *p* = 0.64). Only hypertension and an eGFR of ≤30 mL/min/1.73 m^2^ predisposed patients to the development of CI-AKI.

Additionally, a higher amount of CM used during complex PCI did not have an impact on developing CM-AKI. In the study by Mukete et al. [16], patients who underwent complex PCI in STEMI did not significantly differ in CI-AKI frequency compared to the infarct-artery revascularization (IRA) group, despite the increased use of CM.

Furthermore, the meta-analysis by Chatterjee et al. [17] confirmed these results. No difference in CI-AKI risk between the complex PCI and IRA-only group was observed (OR: 0.73, 95%CI: 0.34–1.57, *p* = 0.57). In both groups, CI-AKI occurred in less than 2% of patients who underwent PCI. 

In the meta-analysis by Ando et al. [18] including nine studies, radial access was associated with a reduction in CI-AKI risk (OR: 0.57, 95%CI: 0.50–0.66, *p* < 0.0001) compared with femoral access. This observation could be the result of choosing femoral access in patients in a hemodynamically unstable state, burdened with serious comorbidities.

In our study, an increased risk of CI-AKI is related to the CKD presence or history of previous AKI rather than to the CM dose. Some studies suggest that the ratio of CM volume to eGFR could predict the risk of CI-AKI. In the study by Zahler et al. [19], including patients with STEMI treated with PCI, a deterioration in kidney function was observed in 9% of individuals. The CM volume to eGFR ratio was significantly higher in patients with a CI-AKI diagnosis. A cutoff point of ≥2.13 was independently associated with higher CI-AKI frequency in a multivariate logistic regression model. 

Our study revealed that therapy with NSAIDs, as well as loop diuretics, increases the risk of CI-AKI. However, this observation could be the result of the comorbidities in the patients treated with those groups of drugs, rather than the pharmacotherapy itself. Only a few previous studies address this problem, but the results are not compatible with each other [20,21]. 

Only a few publications mention CI-AKI in patients who have received CM more than once during hospitalization. In the study by Gu et al. [20], CI-AKI occurred more frequently in patients who received a second dose of CM in a short period of time (12.4% for 1–3 days vs. 5.0% for 4–6 days, *p =* 0.008). In our study, sudden cardiac arrest was related to the higher CI-AKI frequency, while, in the cited study, intra-aortic balloon pump usage and treatment with diuretics were the independent risk factors for kidney function deterioration.

Furthermore, also in the study by Park et al. [22], a short interval between the PCI and contrast-enhanced computed tomography was the independent risk factor for CI-AKI (≤2 days vs. >14 days: HR *=* 2.37, 95%CI: 1.105–5.098, *p =* 0.018; 3–14 days vs. >14 days: HR = 2.07, 95%CI: 0.960–4.445, *p* = 0.064).

### Study Limitations

The limited number of enrolled patients and the single-center, retrospective character of this study were the main limitations. Only the association, but not the “cause-and-effect” relationship, between CM-AKI and the amount of CM used could be investigated, due to the characteristics of the study. Patients with CKD were not excluded from the study, and information on patients’ hydration status was not collected. Also, long-term follow-up was not provided. The avoidance of performing subsequent procedures with the use of CM in patients with a high risk of AKI may have an impact on the study results. Further multicenter studies are required to explore that issue. 

## 5. Conclusions

In patients exposed to repeated doses of CM, the risk of AKI is related to the known risk factors for kidney function deterioration, rather than to the total volume of CM.

In the study, the association between multiple exposures to CM and the risk of developing AKI was not confirmed. 

## Figures and Tables

**Figure 1 jcm-12-02166-f001:**
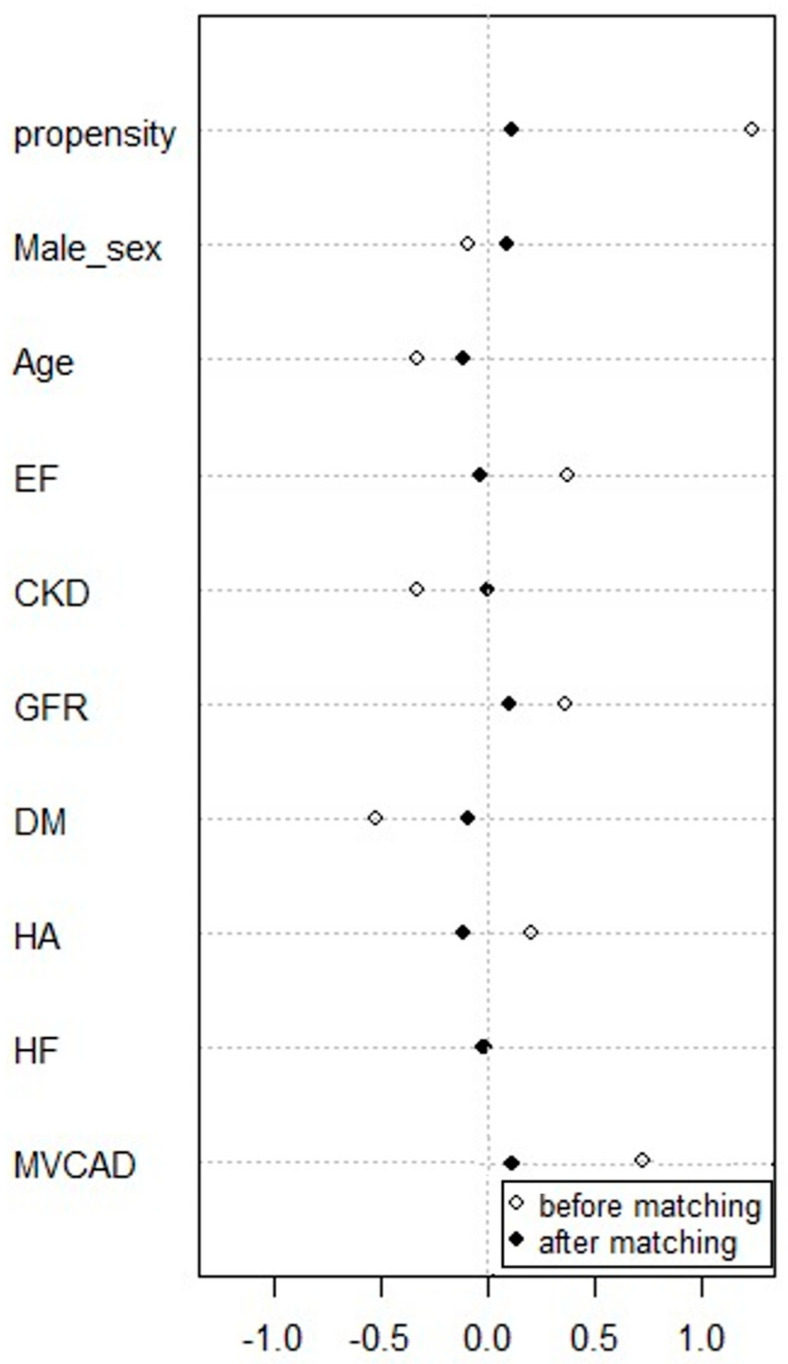
Propensity score matching of study and control groups. EF—left ventricular ejection fraction; CKD—chronic kidney disease; GFR—estimated glomerular filtration rate; DM—diabetes mellitus; HA—arterial hypertension; HF—heart failure; MVCAD—multivessel coronary artery disease.

**Figure 2 jcm-12-02166-f002:**
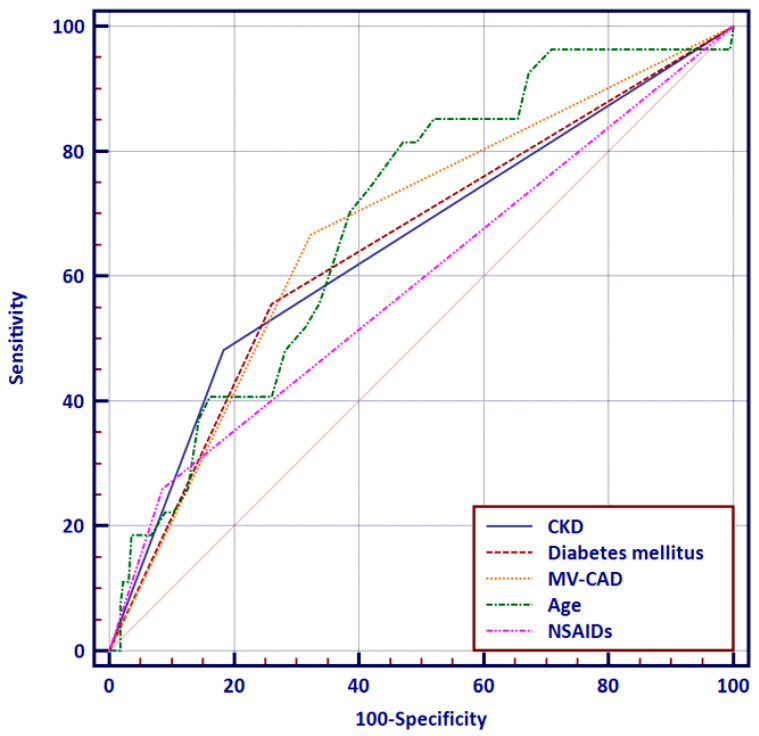
Receiver operating characteristic curve of independent predictors of CI-AKI onset. CKD—chronic kidney disease; MV-CAD—multivessel coronary artery disease; NSAIDs—non-steroidal anti-inflammatory drugs.

**Table 1 jcm-12-02166-t001:** Demographic and clinical characteristics in the overall population and study and control groups.

Variable	Overall Population *n* = 291	Study Group *n* = 138	Control Group *n* = 153	*p*-Value
Age [years]	67.59 ± 11.59	69.57 ± 11.72	65.82 ± 11.21	0.006
BMI [kg/m^2^]	28.47 ± 4.48	28.25 ± 4.40	28.63 ± 4.54	0.377
Male sex	183 (62.9%)	90 (65.2%)	93 (60.8%)	0.434
Medical history and pharmacotherapy				
Chronic kidney disease	71 (24.5%)	43 (31.4%)	28 (18.3%)	0.010
History of AKI	8 (2.8%)	3 (2.2%)	5 (3.3%)	0.562
History of nephrectomy	7 (2.41%)	3 (2.2%)	4 (2.6%)	0.800
History of MI	89 (30.7%)	29 (21.0%)	60 (39.5%)	0.001
History of stroke	26 (9.0%)	16 (11.6%)	10 (6.6%)	0.135
History of PCI	99 (34.1%)	31 (22.5%)	68 (44.7%)	0.000
Atrial fibrillation	29 (14.7%)	15 (10.9%)	14 (23.3%)	0.023
Heart failure	107 (36.8%)	51 (37.0%)	56 (36.6%)	0.950
MV-CAD	100 (34.6%)	21 (15.2%)	79 (52.3%)	<0.001
Diabetes	97 (33.5%)	62 (44.9%)	35 (23.0%)	0.000
Arterial hypertension	252 (86.9%)	115 (83.3%)	137 (90.1%)	0.087
Hyperlipidemia	209 (72.1%)	82 (59.4%)	127 (83.6%)	<0.001
Peripheral artery disease	60 (20.8%)	34 (24.6%)	26 (17.3%)	0.127
Cigarette smoking	126 (43.5%)	53 (38.4%)	73 (48.0%)	0.099
COPD	25 (8.6%)	14 (10.1%)	11 (7.2%)	0.378
Statins	150 (61.5%)	41 (42.7%)	109 (73.7%)	<0.001
MRA	52 (17.9%)	25 (18.1%)	27 (17.7%)	0.917
ACEI/ARB	173 (59.5%)	68 (49.3%)	105 (68.6%)	0.001
Thiazide diuretics	27 (10.6%)	9 (8.4%)	18 (12.2%)	0.337
Loop diuretics	63 (24.8%)	34 (32.1%)	29 (19.6%)	0.023
NSAIDs	26 (10.2%)	8 (7.6%)	18 (12.2%)	0.231
Index hospitalization-initial diagnosis and course of treatment				
STEMI	67 (23.0%)	52 (37.7%)	15 (9.8%)	<0.001
NSTE-ACS	169 (58.1%)	59 (42.8%)	110 (71.9%)	
Chronic coronary syndrome	13 (4.5%)	9 (6.5%)	4 (2.6%)	
Aortic valve stenosis	38 (13.1%)	14 (10.1%)	24 (15.7%)	
Aortic aneurysm	3 (1.0%)	3 (2.2%)	0 (0.0%)	
Systolic BP	130 (120; 140)	130 (120; 140)	130 (120; 140)	0.048
Diastolic BP	80 (70; 80)	80 (70; 80)	80 (70; 85)	0.179
ST-segment abnormalities	125 (62.5%)	80 (58.0%)	45 (72.6%)	0.048
Regional wall motion abnormalities	200 (69.0%)	105 (76.1%)	95 (62.5%)	0.012
Proteinuria	35 (12.7%)	20 (15.3%)	15 (10.3%)	0.220
CI-AKI during index hospitalization	36 (12.4%)	18 (13.0%)	18 (11.8%)	0.741
Sudden cardiac arrest during index hospitalization	12 (6.0%)	5 (3.6%)	7 (11.1%)	0.038
In-hospital death	5 (1.75%)	4 (3.0%)	1 (0.7%)	0.128
Laboratory testing				
Troponin [ng/mL]	0.16 (0.02; 1.15)	0.44 (0.07; 1.92)	0.03 (0.01; 0.25)	<0.001
Hemoglobin	13.54 ± 1.67	13.60 ± 1.79	13.50 ± 1.57	0.574
[K+] [mmol/L]	4.27 ± 2.23	4.29 ± 3.03	4.25 ± 1.09	0.121
[Na+] [mmol/L]	140.01 ± 3.30	139.28 ± 3.37	140.67 ± 3.11	<0.001
Urine specific gravity	1.02 ± 0.06	1.02 ± 0.09	1.02 ± 0.01	<0.001
SCr—baseline [mg/dL]	0.94 (0.78; 1.18)	0.95 (0.79; 1.20)	0.92 (0.78; 1.12)	0.439
SCr—24 h [mg/dL]	0.93 (0.76; 1.16)	0.92 (0.75; 1.21)	0.95 (0.77; 1.13)	0.866
SCr—48 h [mg/dL]	1.01 (0.83; 1.21)	0.96 (0.76; 1.24)	1.09 (0.96; 1.19)	0.200
SCr—72 h [mg/dL]	0.96 (0.79; 1.28)	0.93 (0.78; 1.29)	1.05 (0.85; 1.21)	0.649
SCr—7 days	1.07 (0.68; 1.23)	1.09 (0.75; 1.23)	0.78 (0.57; 0.98)	0.149
ΔSCr 24 h—7 days	0.06 (−0.01; 0.17)	0.08 (−0.02; 0.21)	0.05 (0.00; 0.13)	0.355
eGFR—baseline [mL/min/1.73 m^2^]	77.9 (61; 90)	76.0 (56; 90)	78.1 (65; 90)	0.009
Echocardiography				
LVEF [%]	47.70 ± 11.81	45.39 ± 11.63	49.78 ± 11.62	<0.001
IVS [mm]	12.43 ± 3.36	12.92 ± 3.99	12.03 ± 2.71	0.005
ESD [mm]	33.25 ± 8.49	33.39 ± 8.47	33.14 ± 8.54	0.401
EDD [mm]	51.18 ± 7.66	51.02 ± 8.20	51.32 ± 7.19	0.897
LAd [mm]	39.47 ± 6.42	38.86 ± 6.78	39.97 ± 6.07	0.357
Coronary angiography				
Vascular access				
Radial	131 (48.9%)	95 (75.4%)	36 (25.4%)	<0.001
Femoral	118 (44.0%)	30 (23.8%)	88 (62.0%)	
Brachial	19 (7.1%)	1 (0.8%)	18 (12.7%)	
Local vascular complications	15 (7.5%)	12 (8.7%)	3 (4.8%)	0.338
Stenosis > 50% in coronary arteries	223 (78.0%)	125 (91.2%)	98 (65.8%)	<0.001
PCI	188 (65.5%)	113 (82.5%)	75 (50.0%)	<0.001
Intra-aortic balloon pump	6 (2.1%)	6 (4.4%)	0 (0.0%)	0.009
Low-osmolar contrast media	243 (83.8%)	113 (82.5%)	130 (85.0%)	0.566
Volume of contrast—1 [mL]	120 (80; 170)	120 (80; 180)	100 (70; 160)	0.114
Volume of contrast—2 [mL]	120 (100; 160)	120 (100; 160)	-	-
Volume of contrast—3 [mL]	100 (60; 130)	100 (60; 130)	-	-
Total volume of contrast	180 (100; 270)	265.00 (200; 330)	120.00 (80; 160)	<0.001
Time between contrast media dose [days]	2.00 (2.00; 4.00)	2.00 (2.00; 4.00)	-	-

AKI—acute kidney injury; BMI—body mass index; BP—blood pressure; COPD—chronic obstructive pulmonary disease; eGFR—estimated glomerular filtration rate; MI—myocardial infarction; MV-CAD—multivessel coronary artery disease; MRA—mineralocorticoid receptor antagonist; NSAIDs—non-steroidal anti-inflammatory drugs; SCr—serum creatinine concentration; LAd—left atrial diameter; EDD—end-diastolic diameter; ESD—end-systolic diameter; LVEF—left ventricular ejection fraction; PCI—percutaneous coronary intervention; STEMI—ST-elevation myocardial infarction; NSTE-ACS—non-ST-elevation acute coronary syndrome.

**Table 2 jcm-12-02166-t002:** Demographic and clinical characteristics of non-CI-AKI and CI-AKI groups.

Variable	Non-CI-AKI *n* = 255	CI-AKI Group *n* = 36	*p*-Value
Age [years]	66 ± 11.5	73 ± 10.8	0.001
BMI [kg/m^2^]	28.5 ± 4.5	28 ± 4.3	0.609
Male sex	161 (63%)	22 (61%)	0.814
Medical history and pharmacotherapy			
Chronic kidney disease	53 (20.9%)	18 (50%)	<0.001
History of AKI	5 (1.97%)	3 (8.33%)	0.029
History of nephrectomy	5 (1.97%)	2 (5.6%)	0.189
History of MI	71 (27.9%)	18 (50%)	0.007
History of stroke	20 (7.87%)	6 (16.7%)	0.084
History of PCI	83 (32.7%)	16 (44.4%)	0.502
Atrial fibrillation	33 (13%)	9 (25%)	0.074
Heart failure	90 (35.3%)	17 (47.2%)	0.165
MV-CAD	78 (30.7%)	22 (62.9%)	<0.001
Diabetes	77 (30.3%)	20 (55.6%)	0.003
Arterial hypertension	221 (87%)	311 (86.1%)	0.881
Hyperlipidemia	182 (71.7%)	27 (75%)	0.675
Peripheral artery disease	47 (18.7%)	13 (36.1%)	0.016
Cigarette smoking	113 (44.5%)	13 (36.1%)	0.343
COPD	22 (8.7%)	3 (8.3%)	0.948
Statins	131 (60.7%)	19 (67.9%)	0.461
MRA	46 (18%)	6 (16.7%)	0.841
ACEI/ARB	152 (59.6%)	21 (58.3%)	0.884
Thiazide diuretics	26 (11.5%)	1 (3.45%)	0.184
Loop diuretics	52 (23%)	11 (39.3%)	0.06
NSAIDs	19 (8.4%)	7 (25%)	0.006
Index hospitalization-initial diagnosis and course of treatment			
STEMI	60 (23.5%)	7 (19.4%)	0.934
NSTE-ACS	48 (58%)	21 (58.3%)	
Chronic coronary syndrome	11 (4.3%)	2 (5.6%)	
Aortic valve stenosis	32 (12.6%)	6 (16.7%)	
Aortic aneurysm	3 (1.2%)	0 (0%)	
Systolic BP	130 (120; 140)	120 (110; 140)	0.160
Diastolic BP	80 (70; 80)	80 (70; 80)	0.376
ST-segment abnormalities	109 (63.4%)	16 (57.1%)	0.528
Regional wall motion abnormalities	173 (68.1%)	27 (75%)	0.403
Proteinuria	30 (12.2%)	5 (16.1%)	0.540
Sudden cardiac arrest during index hospitalization	8 (4.6%)	4 (14.3%)	0.045
In-hospital death	4 (1.6%)	1 (2.9%)	0.596
Laboratory testing			
Troponin [ng/mL]	0.15 (0.02; 1.0)	0.49 (0.03; 1.5)	0.190
Hemoglobin	13.7 ± 1.63	12.8 ± 1.8	0.008
[K+] [mmol/L]	4.3 ± 2.4	4.3 ± 0.6	0.083
[Na+] [mmol/L]	140.2 ± 3.2	138.6 ± 3.6	0.009
Urine specific gravity	1.03 ± 0.01	0.99 ± 0.18	0.618
SCr—baseline [mg/dL]	0.93 (0.77; 1.2)	1 (0.8; 1.2)	0.133
SCr—24 h [mg/dL]	0.9 (0.75; 1.1)	1.06 (0.8; 1.3)	0.013
SCr—48 h [mg/dL]	0.98 (0.8; 1.1)	1.42 (1.1; 1.9)	0.001
SCr—72 h [mg/dL]	0.93 (0.8; 1.2)	1.59 (0.9; 1.8)	0.003
SCr—7 days	0.98 (0.6; 1.1)	1.44 (1; 2.1)	0.05
ΔSCr 24 h—7 days	0.05 (−0.03; 0.1)	0.46 (0.4; 0.7)	<0.001
eGFR—baseline [mL/min/1.73 m^2^]	78 (63.7; 90)	66.5 (49; 90)	0.057
Echocardiography			
LVEF [%]	48 ± 11	42 ± 14	0.017
IVS [mm]	12 ± 3	12.5 ± 3	0.206
ESD [mm]	33 ± 7.8	35 ± 12.3	0.261
EDD [mm]	51 ± 6.9	52 ± 12	0.127
LAd [mm]	39.2 ± 6	41.2 ± 8.9	0.007
Coronary angiography			
Vascular access			
Radial	116 (49.2%)	15 (46.9%)	0.936
Femoral	103 (43.6%)	15 (46.9%)	
Brachial	17 (7.2%)	2 (6.3%)	
Local vascular complications	11 (6.4%)	4 (14.8%)	0.121
Stenosis > 50% in coronary arteries	190 (75.7%)	25 (94.3%)	0.663
PCI	166 (65.9%)	22 (62.9%)	0.725
Intra-aortic balloon pump	6 (2.4%)	0 (0%)	0.351
Low-osmolar contrast media	215 (84.7%)	28 (77.8%)	0.295
Volume of contrast—1 [mL]	120 (80; 160)	130 (80; 230)	0.201
Volume of contrast—2 [mL]	125 (100; 170)	100 (80; 150)	0.093
Volume of contrast—3 [mL]	90 (50; 125)	200 (200; 200)	0.121
Total volume of contrast	180 (100; 270)	205 (100; 375)	0.417
Time between contrast media dose [days]	2 (2; 4)	3 (2; 3)	0.203

AKI—acute kidney injury; BMI—body mass index; BP—blood pressure; COPD—chronic obstructive pulmonary disease; eGFR—estimated glomerular filtration rate; MI—myocardial infarction; MV-CAD—multivessel coronary artery disease; MRA—mineralocorticoid receptor antagonist; NSAIDs—non-steroidal anti-inflammatory drugs; SCr—serum creatinine concentration; LAd—left atrial diameter; EDD—end-diastolic diameter; ESD—end-systolic diameter; LVEF—left ventricular ejection fraction; PCI—percutaneous coronary intervention; STEMI—ST-elevation myocardial infarction; NSTE-ACS—non-ST-elevation acute coronary syndrome.

**Table 3 jcm-12-02166-t003:** Univariable and multivariable logistic regression analysis of different predictors of the onset of contrast-induced acute kidney injury.

Univariable Analysis			
Variable	OR	95%CI	*p*-Value
Study group (double contrast administration)	0.8889	0.4423–1.7866	0.7409
>50% stenosis in coronary artery	0.6721	0.0632–7.1511	0.7419
CKD	7.9571	1.8037–35.1031	0.0062
Diabetes	4.0328	1.1748–13.8442	0.0267
History of AKI	7.5117	0.6943–81.2714	0.097
MV-CAD	22.6315	3.9675–129.0946	0.0004
Age	1.0787	1.0028–1.1605	0.0419
NSAIDs use	6.7687	1.4828–30.8975	0.0136
PAD	4.4525	1.2313–16.1007	0.0228
Loop diuretics	0.189	0.0391–0.9128	0.0381
LVEF	0.9886	0.9396–1.0402	0.6596
Total volume of contrast	0.9921	0.9778–1.0066	0.2828
Total volume of contrast-to-body weight ratio	2.3129	0.7715–6.9335	0.1344
**Multivariable Logistic Regression**			
**Variable**	**logOR**	**95%CI**	***p*-Value**
CKD	3.7268	1.0732–12.9415	0.0383
Diabetes	3.0242	1.0340–8.8446	0.0433
MV-CAD	9.4553	2.6483–33.7591	0.0005
Age [years]	1.0738	1.0073–1.1447	0.029
NSAIDs use	6.4447	1.7891–23.2150	0.0044

CKD—chronic kidney disease; AKI—acute kidney injury; MV-CAD—multivessel coronary artery disease; NSAIDs—non-steroidal anti-inflammatory drugs; PAD—peripheral artery disease; LVEF—left ventricular ejection fraction.

## Data Availability

The data are available upon request from the corresponding author.

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
