# Peer review of "Repeated Dose of Contrast Media and the Risk of Contrast-Induced Acute Kidney Injury in a Broad Population of Patients Hospitalized in Cardiology Department"

_jcm, 2023, doi:10.3390/jcm12062166_

Round 1

Reviewer 1 Report

The submitted manuscript brings up an important topic and provides relevant results. It is scientifically sound, clear, and presented in a well-structured manner. The paper is relevant for the field. However, one small remark still remains.   The authors write in Conclusion that “a causal-and-effect relationship between exposures to CM and the risk of developing AKI has not been confirmed”. This is not entirely accurate. The causal relationships can only be confirmed in a longitudinal study. In a cross-sectional study, which is the present in this article, only associations can be assessed; causal relationships cannot be studied at all. Therefore, the statement about “a causal-and-effect relationship” refers more to the Study limitations than to the Conclusion. Herewith, this is not about non-confirming causal relationships, but about the impossibility of studying them in this work.   

Author Response

Reviewer 1

The submitted manuscript brings up an important topic and provides relevant results. It is scientifically sound, clear, and presented in a well-structured manner. The paper is relevant for the field. However, one small remark still remains.   The authors write in Conclusion that “a causal-and-effect relationship between exposures to CM and the risk of developing AKI has not been confirmed”. This is not entirely accurate. The causal relationships can only be confirmed in a longitudinal study. In a cross-sectional study, which is the present in this article, only associations can be assessed; causal relationships cannot be studied at all. Therefore, the statement about “a causal-and-effect relationship” refers more to the Study limitations than to the Conclusion. Herewith, this is not about non-confirming causal relationships, but about the impossibility of studying them in this work.

Thank You for Your insight. According to Your suggestions we rewrote the Conclusions (“In the study the associationbetween multiple exposures to CM and the risk of developing AKI was not confirmed.” Page 21, Line 377-378).

As You have underlined, the causal relationship cannot be proven in cross-sectional study. We added that information in the Study limitations (“Only the association, but not the “causal-and-effect” relationship between CI-AKI and amount of used CM could the investigated due to the character of the study.” Page 20, Line 366-368).

Reviewer 2 Report

The topic is interesting, dealing with a common complication in cardiological patients undergoing contrast media injection. The conclusions are that contrast media multiple use during the same journey does not predict AKI. However the study is biased by a small sample size: can the author provide the sample size calculation in order to know if their sample size is adequate to answer the research question? In addition, what is the novelty of their study with respect to previous ones.

Minor consideration: how many patients were pre-treated by N-acetyl-cisteine?

Author Response

Reviewer 2

The topic is interesting, dealing with a common complication in cardiological patients undergoing contrast media injection. The conclusions are that contrast media multiple use during the same journey does not predict AKI. However the study is biased by a small sample size: can the author provide the sample size calculation in order to know if their sample size is adequate to answer the research question? In addition, what is the novelty of their study with respect to previous ones.

Thank you for raising that issue. The present study is unique and only one study before has directly evaluated the rate of CI-AKI in patients subject to repeated contrast administration [20]. This cited study did not comprise control group, thus we don’t have input value of projected CI-AKI incidence in study and control group. Thus, it was virtually impossible to calculate the sample size before the study processing. Still, if we assume that the incidence of CI-AKI in case of single dose of contrast media was 4.3% based on study by Mehran et al and that the incidence of CI-AKI was 9.2% in patients with double contrast dose [20], the overall sample size of the study for dichotomous variable with 80% power should be equal to 820 (n=410 in both groups).

Dichotomous Endpoint, Two Independent Sample Study

Sample Size

Group 1

410

Group 2

410

Total

820

Study Parameters

Incidence, group 1

4.3%

Incidence, group 2

9.2%

Alpha

0.05

Beta

0.2

Power

0.8

The advantage of our study over previous studies is that it proved the evidence of many
CI-AKI risk factors. The study revealed that, age, CKD, diabetes, MV-CAD, PAD, NSAIDs and Loop diuretics usage had an impact on the CI-AKI development.

Minor consideration: how many patients were pre-treated by N-acetyl-cisteine?

Thank You for that question. None of the patients were pre-treated by N-acetyl-cisteine. According to the ESC Guidelines on Myocardial Revascularization we provide only adequate fluid supply before procedures with CM usage and use routine high-dose statin pretreatment.

Reviewer 3 Report

I would like to congratulate authors for their results in presented manuscript entitled: “Repeated dose of contrast media and the risk of contrast-induced acute kidney injury in a broad population of patients hospitalized in cardiology department”. They aimed to evaluate the impact of repeated contrast media (CM) administration during single hospitalization on the rate of CI-AKI. Authors presented interesting results with great potential, however, I have comments on this:

Major:

The main and the significant shortcoming of the study are totally unadjusted groups of patients. Authors should plan the study using propensity score matching as they use retrospective data for the analysis. Otherwise, their results are incorrect.

Minor:

1. Authors should state the objective of the study in the main text.

 2. There is no need to duplicate group allocations of the patients in methods and results sections.

Author Response

Reviewer 3

I would like to congratulate authors for their results in presented manuscript entitled: “Repeated dose of contrast media and the risk of contrast-induced acute kidney injury in a broad population of patients hospitalized in cardiology department”. They aimed to evaluate the impact of repeated contrast media (CM) administration during single hospitalization on the rate of CI-AKI. Authors presented interesting results with great potential, however, I have comments on this:

Major:

The main and the significant shortcoming of the study are totally unadjusted groups of patients. Authors should plan the study using propensity score matching as they use retrospective data for the analysis. Otherwise, their results are incorrect.

Thank You for that comment. Although the CI-AKI problem was the topic of many previous studies, we still don’t have enough data on risk of multiple CM usage. We must admit that the study population was heterogenous. Using propensity score matching in that population would lead to rejection of many included patiens. It would result in receiving too small study groups to analyze the problem. Despite the flaws, we still think that problem of CI-AKI requires more investigation, and our study is valuable due to its innovative character. The study represents a real-world registry and, despite higher prevalence of comorbidities in repeated CM dose group, the rate of CI-AKI was comparable in both cohorts. Although the study is too small to draw definite conclusions, it suggests that patients requiring repeated coronary angiography or coronary angiography and CT are not prone to development of CI-AKI to much greater extent than in patients with single dose of CM. We added that issue in the Study limitations (Page 20, Line 366).

Minor:

  1. Authors should state the objective of the study in the main text.

That You for Your comment- the aim of the study is described in the Intoduction (Page 4, line 75-77).

  1. There is no need to duplicate group allocations of the patients in methods and results sections.

Thank You for Your insight, we decided to present group allocation only in the Methods section. 

Round 2

Reviewer 2 Report

accept

Author Response

Thank You for Your suggestions.

Reviewer 3 Report

I am pleased that authors paid attention to raised concerns in the previous manuscript review and made some corrections. Undoubtedly, the problem of CI-AKI needs to be analyzed extensively and I admit that authors attempt is a good contribution to the issue. However, despite the indication the main flaws of the article in the limitation section I still feel that the valuable of the study is not as high as authors could present when they will perform the propensity score matching analysis. To my opinion, it is a better way to improve the manuscript. A real-world character of the study is a good but not scientific way in the investigation. Of course, using propensity score matching in the population would lead to rejection of many included patients. I suggest that insufficient number of patients may be compensate by the multicenter study. Subsequently the scientific power of the study would be way more stronger.

Author Response

am pleased that authors paid attention to raised concerns in the previous manuscript review and made some corrections. Undoubtedly, the problem of CI-AKI needs to be analyzed extensively and I admit that authors attempt is a good contribution to the issue. However, despite the indication the main flaws of the article in the limitation section I still feel that the valuable of the study is not as high as authors could present when they will perform the propensity score matching analysis. To my opinion, it is a better way to improve the manuscript. A real-world character of the study is a good but not scientific way in the investigation. Of course, using propensity score matching in the population would lead to rejection of many included patients. I suggest that insufficient number of patients may be compensate by the multicenter study. Subsequently the scientific power of the study would be way more stronger.

Thank You for Your insight. According to Yours suggestions we decided to analyze our data using propensity score matching taking into consideration the following set of variables: age, sex, left ventricular ejection fraction (LVEF), chronic kidney disease (CKD), estimated glomerular filtration rate (eGFR), prevalence of arterial hypertension, diabetes mellitus, heart failure and multivessel coronary artery disease. After the propensity score matching the rate of CI-AKI was comparable in study and control group (13.1% vs. 13.1%, P=1.0), which is compatible with our previous analysis. We have updated paragraphs: Abstract (page 2, lines 32-34) and Statistical analysis (page 5, lines 113-119). We also added paragraph Propensity score matching: study vs. control group (page 10-11, lines 180-200).

Thank You once more for Your invaluable help to improve our research.